# Ensemble forecast of human West Nile virus cases and mosquito infection rates

Nicholas B. DeFelice[1], Eliza Little[1], Scott R. Campbell[2] & Jeffrey Shaman[1]

West Nile virus (WNV) is now endemic in the continental United States; however, our ability to predict spillover transmission risk and human WNV cases remains limited. Here we develop a model depicting WNV transmission dynamics, which we optimize using a data assimilation method and two observed data streams, mosquito infection rates and reported human WNV cases. The coupled model-inference framework is then used to generate retrospective ensemble forecasts of historical WNV outbreaks in Long Island, New York for 2001–2014. Accurate forecasts of mosquito infection rates are generated before peak infection, and > 65% of forecasts accurately predict seasonal total human WNV cases up to 9 weeks before the past reported case. This work provides the foundation for implementation of a statistically rigorous system for real-time forecast of seasonal outbreaks of WNV.

[1] Department of Environmental Health Sciences, Mailman School of Public Health, Columbia University, New York, New York 10032, USA. [2] Arthropod-Borne Disease Laboratory, Suffolk County Department of Health Services, Yaphank, New York 11980, USA. Correspondence and requests for materials should be addressed to N.B.D. (email: nbd2113@columbia.edu).

West Nile virus (family *Flaviviridae*, genus *Flavivirus*, WNV) was first identified in North America in New York City during the summer of 1999 (ref. 1) and by 2003 had spread throughout the continent and established itself as the leading cause of domestically acquired arthropod-borne viral (arboviral) disease in the United States[2,3]. While it is estimated that most infections of WNV are asymptomatic, 20–30% develop acute systemic febrile illness and <1% experience neuroinvasive disease (for example, meningitis, encephalitis or myelitis)[4,5]. In 2012, human cases of WNV surged to numbers not seen since 2003 suggesting that it will continue to produce unpredictable local and regional outbreaks throughout the US[6]. Although, WNV outbreaks recur annually, at present our ability to predict the timing, magnitude and duration of local WNV outbreaks remains limited.

Currently, with no vaccine or specific treatment for WNV, the primary defenses against an outbreak are personal protective behaviours (for example, mosquito repellent) and community-based mosquito control programs[6,7]. On the population level, community-based mosquito control programs are the most effective tool to prevent the spread of WNV[6]. However, these programs are typically inadequately funded[8] and the effectiveness of these control measures can be difficult to assess due to naturally occurring confounding factors, such as host-vector interaction and susceptibility of host species[9–13]. These confounding factors result in seasonal outbreaks that vary in size and scope, and where, even after an outbreak has begun, it remains difficult to predict the future characteristics of the epidemic curve[14–16]. If outbreak characteristics could be reliably forecast, public health response efforts might be better coordinated and mosquito control programs could improve the use of these limited resources. Such a forecast system could also improve our understanding of the epidemiology, ecology and risk factors critical for controlling an outbreak.

Recently, a number of model-inference frameworks have been developed and used to generate accurate ensemble forecasts of infectious diseases, such as influenza and Ebola[17–21]. These forecasting frameworks consist of three components: an epidemiological model, surveillance data and a data assimilation method that bridges the model output and surveillance data.

Here we extend the above approach to the prediction of WNV. We first develop and validate a compartmental model that describes the zoonotic transmission of WNV between mosquito vectors and avian hosts while also accounting for spillover transmission to humans. The model is then coupled with two observed data streams—mosquito infection rates and reported human cases of WNV during 2001–2014 in Suffolk County, New York—using the ensemble adjusted Kalman filter (EAKF)[22] for data assimilation. Weekly retrospective forecasts of WNV are generated for the 2001 to 2014 seasons using this coupled model-EAKF framework. The findings indicate that accurate forecasts of WNV outcomes can be generated with considerable lead-time, and provide a foundation for a statistically rigorous system for real-time forecast of seasonal outbreaks of WNV.

## Results

**Association of human WNV cases and mosquito infection rates.** In Suffolk County the seasonal sum of weekly observed infected mosquito proportions is strongly correlated with the total number of human WNV cases ($r = 0.76$, $P = 0.002$, Fig. 1); weekly human cases were lag correlated with mosquito infection rates during the prior week (Supplementary Fig. 1). In contrast, the number of mosquitoes observed and mosquitoes caught per trap night had no correlation to the number of infected humans. These findings indicate that accurate prediction of infectious

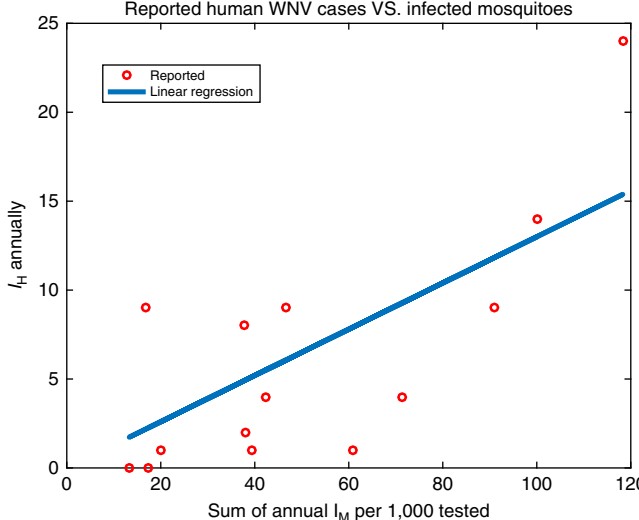

**Figure 1 | Scatterplot of the annual total number of human WNV cases in Suffolk County, NY, USA as a function of the annual sum of weekly observed mosquito WNV infection rates.** The two quantities are highly linearly correlated ($r = 0.76$, $P = 0.002$).

mosquito rates throughout the season has the potential to inform reliable prediction of total human WNV cases.

**The WNV transmission model.** We developed a compartmental model that depicts the transmission dynamics of WNV among mosquitoes and birds, as well as spillover transmission to humans, and used this model to forecast the number of human WNV cases along with the peak timing, peak magnitude and total number of infectious mosquitoes over a season. The compartmental WNV model includes five state variables representing susceptible mosquitoes and birds and infected mosquitoes, birds and humans. Susceptible humans and recovered birds are represented implicitly. All populations are assumed to be constant and the mosquito population is assumed to be female and actively seeking a blood meal (see Supplementary Methods).

Free simulation with the compartmental model captures the general shape of an outbreak in Suffolk County (Supplementary Fig. 2). WNV transmission dynamics among birds and mosquitoes are represented by three parameters: (1) the life expectancy of the mosquito, (2) the recovery time for an infected bird and (3) the contact rate between mosquitoes and birds. While two of these parameters are relatively constant during the mosquito season, the contact rate changes due to changes in mosquito feeding preferences. Many mosquito species, including the dominant WNV vector of Suffolk County, *Culex pipiens*, transition from preferential feeding on birds to mammals over the course of a season[23–26]. Consequently, we depicted contact between vector mosquitoes and avian hosts using a logistic function that represents this change in mosquito biting preference (Supplementary Fig. 2). Accounting for this transition in feeding preference allows the model to capture the temporal change in the contact rate between mosquitoes and birds, which is important in predicting future mosquito infection rates.

Next, we coupled the model with the EAKF, which we used for model optimization and parameter estimation[17,22]. We used output from model free simulation as an initial target to test the optimization efficiency of the filter (Fig. 2). The model state variables and parameters from the free simulation were defined as the 'truth,' and synthetic observations were generated through the addition of noise to that truth. The model-EAKF system not only simulated the 'true' state variables well (infectious mosquitoes, $I_M$,

and human WNV cases, $I_H$) but, in addition, also inferred the unobserved state variables and epidemiologically significant parameters that help define the number of human WNV cases, along with the peak timing, magnitude and duration of infectious mosquitoes during an outbreak (Supplementary Figs 3–6). These inferences included estimation of the parameters

defining mosquito-to-human spillover transmission rates, $\eta$, and mosquito-bird transmission rates, $A$, $r$ and $\beta(t)$. For more information on this validation of the model-EAKF system, see the Supplementary Information.

**Retrospective forecast of human and mosquito WNV infection.** We next used the combined model-EAKF system to generate retrospective forecasts of infectious mosquito rates and human WNV cases. For each annual outbreak during 2001–2014, the model-EAKF system was initiated 4 weeks before the first positive mosquito observation. Each week, observations of mosquito infection rates and human WNV cases were assimilated using the EAKF, and a forecast was generated by integrating the posterior model ensemble to the end of the outbreak season. Weekly forecasts were produced from the first detection of infectious mosquitoes to the end of the season. Figure 3 shows successive forecasts of infectious mosquitoes and human cases during the 2010 season, beginning 4 weeks before peak mosquito infection rate until 2 weeks past that peak. The ensemble forecasts capture the overall structure of the outbreak among mosquitoes well in advance of peak infection; further, predictions of human WNV cases come in line with observed values as more observations are assimilated.

Seasonal forecast accuracy was examined for four metrics: total human WNV cases, total infectious mosquitoes, peak infectious mosquitoes and peak timing. Forecasts were deemed accurate if the ensemble mean trajectory was within $\pm 25\%$ or $\pm 1$ case, whichever was larger, of the first metric, within $\pm 25\%$ of the next two metrics, and within $\pm 1$ week of the fourth metric. Supplementary Figs 7–10 present this forecast accuracy across all seasons (2001–2014) as a function of calendar week. Forecasts

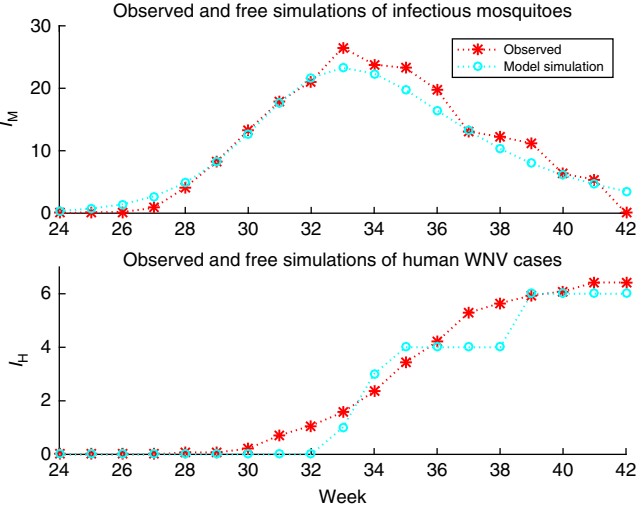

**Figure 2 | Average weekly observations and free model simulation of infectious mosquito rates and human WNV cases.** Only 25% of human cases had been reported by week 33, the week that average observations of infectious mosquitoes peaked. Average weekly observations (2001-2014, red *) and free model simulation (cyan o).

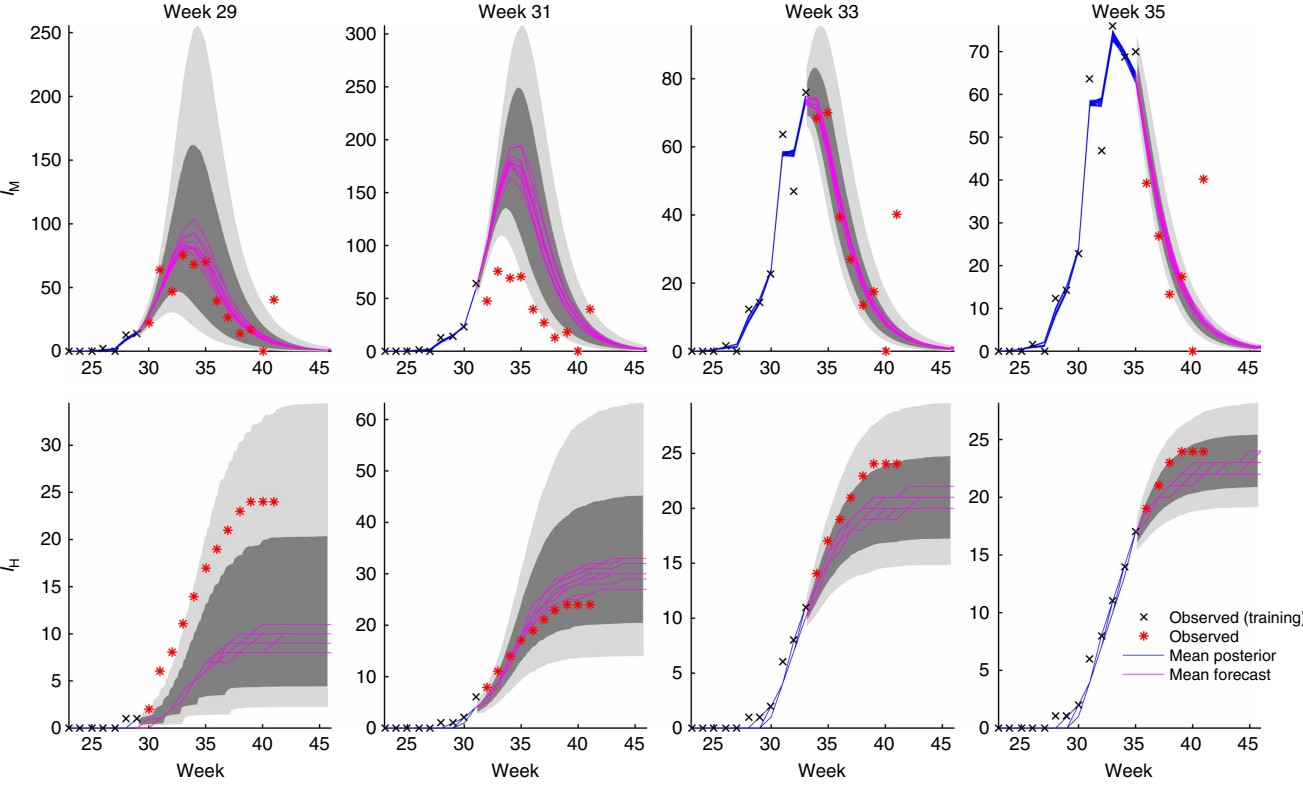

**Figure 3 | Ten bi-weekly forecasts of infectious mosquitoes and human WNV cases for 2010.** The magenta lines are the ensemble mean forecasts, the grey area is the spread of the ensemble forecast (light grey represents area between the 10th and 90th percentile and the darker grey area represents the spread between the 25th and 75th percentile), blue lines are the ensemble mean posterior distribution, black *x*'s are data points assimilated into the model and red * are future observations.

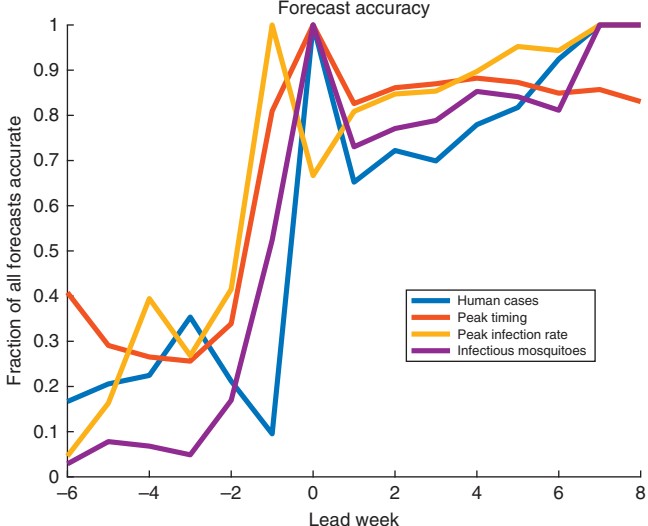

**Figure 4 | Results for 2001–2014 retrospective forecasts.** Shown are the fraction of forecasts accurate as a function of lead week for the metrics human WNV cases (blue), peak timing (week of peak mosquito infection rates, orange), peak infection rate (yellow) and total infectious mosquitoes (purple). A forecast was deemed accurate if: (1) peak timing was within ±1 week of the observed peak of infectious mosquitoes; (2) peak infection rate was within ±25% of the observed peak infection rate; (3) total infectious mosquitoes were within ±25% of the observed; and (4) human WNV cases were within ±25% or ±1 case of the total number of reported cases, whichever was larger. Note that for all metrics lead week is shown with respect to the week of peak mosquito infection.

were further grouped by prediction lead-time, here defined as the week of forecast generation minus the week of predicted peak mosquito infection (Fig. 4). Forecasts of peak timing were accurate >80% of the time with 1 week prediction lead. Forecasts of peak infectious mosquito number were also accurate 1 week before the peak and were >67% accurate when the predicted lead-time was at the peak or past the peak. For the total number of infected mosquitoes 52% of forecasts were accurate one week before the predicted peak, and >73% were accurate at or past the predicted peak.

Accurate estimation and prediction of infectious mosquito numbers and EAKF optimization of spillover transmission rates enabled accurate forecast of total human WNV cases. At 0, 1 and 2 weeks past peak predicted mosquito infection rates, the forecasts of total human WNV cases were accurate 100, 65 and 72% of the time, respectively. On average, only one-fifth of human cases are reported before the week of peak mosquito infection. Consequently over the season, the forecast of human cases, near the predicted peak of mosquito infection, is before the reporting of the majority of human cases.

## Discussion

Our findings demonstrate that a compartmental model of WNV, iteratively optimized with data assimilation methods and weekly observations of mosquito infection rates and human WNV cases, can produce accurate forecasts of mosquito infection rates, infectious biting pressure and human cases. With timely provision of these data, real-time operational forecasts can be generated. Such information has the potential to help public health officials, mosquito control programs and parks departments target control of infectious mosquito populations, alert the public when WNV spillover transmission risk is elevated, and determine if parks and camping grounds should be closed.

The forecasts provide ample lead-time for undertaking targeted interventions. Infectious mosquito peak timing, an important indicator of spillover transmission potential, is forecast accurately up to 6 weeks in advance but with high accuracy 1 week in advance (Fig. 4). Indeed, the onset of human WNV cases was within 1 week of the observed infectious mosquito peak in half of the years with human cases, and only 3 years reported cases earlier. In addition, by the week of the predicted peak, the model-EAKF system has undergone extensive training and begins to forecast other characteristics more accurately, with greater than 67%, 73% and 65% accuracy pertaining to the peak infection rate of mosquitoes, the seasonal total of infectious mosquitoes, and the total number of human WNV cases reported during the outbreak, respectively.

Observed total infected mosquito proportions are strongly correlated with the total number of human WNV cases (Fig. 1). This association, coupled with sufficient optimization of the parameter $\eta$ determining mosquito-to-human transmission rates (equation 5), allows accurate forecast of total human WNV cases with considerable lead. Indeed accurate forecasts of total human WNV cases were generated up to 9 weeks before the end of an outbreak, Supplementary Fig. 11.

There were some differences in forecast accuracy among high and low years. During high case years, which we define as years with more than 6 human WNV cases, infectious mosquito forecasts were more accurate than forecasts of human cases, Supplementary Fig. 12. The tendency, in some high case years, was to underestimate human WNV cases until six cases had been observed; however, during low case years the system tended to predict a high number of cases early in the season. Consequently, the current forecast system has a proclivity, early in the season, to commit a type II error, for example, forecast a low human case year when it is really a high year. As the season progresses and more cases are observed, the system adjusts and the forecasts improve so that total human WNV cases are accurately forecast with substantial lead times (2–9 weeks before the end of the season).

As more years of data become available, we hope to further validate and refine estimates of the relationship between peak timing of infectious mosquitoes, total numbers of infected mosquitoes, and spillover infection of humans. We also hope to entrain environmental variables, such as temperature, into the core model structure, to explore whether observations of those variables help further constrain model dynamics and forecast of the timing and magnitude of infectious mosquitoes and potential risk to humans.

In building this WNV forecast system, we had to choose which processes to include and which to exclude in the core dynamic model. A number of unrepresented effects, including ongoing mosquito control efforts, within county spatial heterogeneity[10], bird migration[13], variable susceptibility among different hosts[9] and vectors[11], WNV strain variability[12], vertical transmission[11], mosquito overwintering patterns[27] and the extrinsic incubation period[28], may affect WNV transmission dynamics and spillover infection to humans. However, inclusion of too many processes results in a high-dimensional model structure, which, given the limited observational data streams available, may be difficult to optimize. On the other hand, a model that is too simple will not contain sufficient dynamics to generate a characteristic WNV outbreak in free simulation and thus will not produce accurate forecasts. The model we chose for the present work is of intermediate complexity. It is capable of producing a realistic outbreak of WNV in free simulation, yet is sufficiently parsimonious to permit state variable and parameter estimation

with the EAKF and accurate forecast of future outcomes given currently available data streams. As more data become available, inclusion of additional effects in the core dynamic model may improve overall system performance.

Though WNV transmission dynamics vary by location[10–13,27,28], the simple forecasting framework presented here was designed for broad application in different settings. To test this generalizability, we applied the model-inference forecast system to one additional location, Cook County Illinois during 2007–2014 (Supplementary Figs 13 and 14). The results from Cook County are consistent with those found for Suffolk County. Human cases were accurately forecast on average 6.3 weeks before the end of an outbreak. Thus, it appears that EAKF optimization allows this relatively simple construct to produce accurate forecasts in locations where WNV, host and vector dynamics differ.

Going forward, it is important to work with mosquito control and public health officials to increase their familiarity with the capabilities and limitations of these forecasts, as well as our own familiarity with potential mosquito interventions and the process for deciding when and where to implement those interventions. By doing so, these forecasts can be presented and interoperated to better support intervention decision and inform the public of potential risks.

## Methods

**Compartmental model.** Our model uses a standard SIR epidemiological framework and is represented by following equations:

$$\frac{dS_M}{dt} = \mu_M N_M - \beta(t)S_M \frac{I_B}{N_B} - \mu_M S_M - \alpha S_M \tag{1}$$

$$\frac{dI_M}{dt} = \beta(t)S_M \frac{I_B}{N_B} - \mu_M I_M + \alpha S_M \tag{2}$$

$$\frac{dS_B}{dt} = -\beta(t)I_M \frac{S_B}{N_B} \tag{3}$$

$$\frac{dI_B}{dt} = \beta(t)I_M \frac{S_B}{N_B} - \frac{I_B}{\delta_B} \tag{4}$$

$$\frac{dI_H}{dt} = Poisson(\eta I_M) \tag{5}$$

where $S_M$ is the number of susceptible mosquitoes, $\mu_M$ is the mosquito birth and death rate, $N_M$ is the mosquito population and is constant over an outbreak, $t$ is time in days, $\beta(t)$ is the contact rate or probability of transmission between birds and mosquitoes at time $t$, $\alpha$ is the rate of WNV seeding into the local model domain before day 200, $I_M$ is the number of infected mosquitoes, $N_B$ is the bird population and is constant over an outbreak, $I_B$ is the number of infected birds, $S_B$ is the number of susceptible birds in the population, $\delta_B$ is the recovery rate of birds, $I_H$ is the number of infected humans, and $\eta$ is a scalar that accounts for the contact rate and probability of transmission from mosquitoes to humans. The probability of WNV spilling over to humans is simulated using a Poisson random number generator.

All model state variables and parameters were estimated simultaneously through EAKF data assimilation. The EAKF is a recursive filtering technique that combines observations with a temporally evolving ensemble of model simulations to generate a posterior estimate of the model state[22]. This process nudges the ensemble mean toward the observations and simultaneously contracts the ensemble variance, and in doing so optimizes the state variables and parameters. The contact rate, $\beta(t)$, was originally specified as a single free parameter, $\beta$; however, we observed during simulation of seasonal WNV that estimates of $\beta$ declined in a characteristic 'S curve' shape (Supplementary fig. 2). This inferred change in the contact rate may be representative of the switch in mosquito feeding preference from avian to mammalian hosts[23,24,26]. We chose to explicitly model this process as a generalized logistic equation:

$$\beta(t) = A + \frac{K - A}{1 + e^{(-r(t - t_0))}} \tag{6}$$

where $A$ is the lower asymptote, $K$ is the upper asymptote, $r$ is the growth rate and $t_0$ is the inflection point. By imposing this form within the model, we can predict future shifts in feeding preference and vector-avian host contact, provided sufficient optimization of the parameters in equation 6.

**Observational data.** Mosquito surveillance in Suffolk County New York was conducted weekly from early June to the middle of October, depending upon the severity of WNV. At the beginning of each season, trap locations were spatially distributed throughout the county and guided by the historical presence of WNV. As the season progressed, mosquito monitoring was expanded within regions where WNV had been identified. Total traps within a season ranged from 47 to 104 traps depending on the year, and the number of traps set each week varied. Approximately half of the traps were operated in or around town, county or state parks. For arboviral analysis, pools were submitted to the New York State Department of Health (Arbovirus Laboratory, Wadsworth Center). WNV analysis was performed by real-time reverse transcription PCR on pools of mosquitoes to determine the presence of WNV. We combined all pools of mosquitoes sampled in a week and used a maximum-likelihood approach to estimate the total weekly proportion of positive mosquitoes (see Supplementary Information for more details on the mosquito data and this calculation)[29].

Weekly human cases of WNV in Suffolk County New York were obtained from ArboNET, the national arboviral surveillance system, from 2001 to 2014 (ref. 30). WNV is a nationally notifiable disease. State and local health departments report the weekly number of human WNV cases to the Centers for Disease Control and Prevention through the ArboNET surveillance system[31]. In this analysis both neuroinvasive and non-neuroinvasive cases were considered (see Supplementary for more detail on human cases of WNV).

**Model-EAKF system.** The EAKF data assimilation method has previously been used in conjunction with a variety of compartmental epidemiological models and infectious disease data to simulate diseases such as influenza and Ebola[17–20]. This data assimilation technique uses Bayes' rule to provide an updated target of the system state at a given point in time, using the current observation and all prior observations. In the process of updating the observed model state variables the EAKF algorithm also adjusts the unobserved state variables and parameters. For further details on how the EAKF adjusts the ensemble prior such that the new moments match the target moments of the posterior predicted by Bayes' theorem see the Supplementary Material and Anderson[22].

In this study, a 300-member ensemble simulation of the SIR compartmental model (equations (1)–(6)) was run in conjunction with the Suffolk County infectious mosquito and human WNV case data and the EAKF. The model-EAKF system contains the modelled state space composed of the five disease state variables and seven parameters $z_t = (S_M, I_M, S_B, I_B, I_H, \mu, A, K, r, t_0, \delta_B,$ and $\eta)$ and the weekly observations of mosquito WNV infection rates and human WNV cases, $y_t = (I_M$ and $I_H)$. Whenever an observation becomes available, in this study observations were reported weekly, the EAKF algorithm assimilates those new observations to update the model observed state variables. The EAKF algorithm also updates the model unobserved state variables and parameters using cross-ensemble co-variability. The model is then integrated forward to the next observation, using the updated (posterior) model state variables and parameters, and the data assimilation updating process is repeated. Through this iterative optimization process, the ensemble of model simulations is better aligned to simulate current local outbreak dynamics. To validate that the EAKF data assimilation optimizes the WNV-compartmental model, we synthetically tested the combined model-EAKF system (see Supplementary Materials).

**Initial conditions of compartmental model.** The compartmental model was initiated with a 300-member ensemble[32] for each outbreak season (June to November) and each ensemble member was initialized with a constant total population $S_{m(0)} = 4,000$, $I_{m(0)} = 0$ $S_{B(0)} = 500$, $I_{B(0)} = 0$ and $I_{H(0)} = 0$. Model parameters were randomly selected from a uniform distribution: $\mu = U(0.05,0.08)$ (refs 33,34), $A = U(0.001,0.015)$, $K = U(0.06,0.1)$, $r = U(-0.2, -0.05)$, $\delta = U(3.8,6.0)$ (ref. 35), $\eta = U(0,0.004)$; and $t_0$ was set 5 to 10 weeks after the timing of the first infectious pool of mosquitoes. Initial priors for mosquito expected lifespan[33,34] and bird duration of infection[35] were selected from the literature whereas initial priors for $\beta(t)$ and $\eta$ were determined during synthetic testing of the model-EAKF system. The simulation was seeded with infected mosquitoes, $\alpha$, during the initial integration period until the middle of July, day 200, at a rate of one infected mosquito per 500,000 susceptible mosquitoes.

**Retrospective forecast.** Retrospective forecasts were generated weekly, the interval of observation, during each outbreak season from 2001 to 2014. For each annual outbreak, the EAKF assimilates weekly observation of mosquito infection rates and human WNV cases from the initiation of an ensemble simulation up to the point of forecast. Each week, forecasts were generated following the most recent update of model state variables and parameters by integrating the WNV compartmental model (equations (1)–(6)) through time until the end of the outbreak. This process was repeated weekly with each successive forecast having one additional week of observational data assimilated. Each ensemble forecast was repeated 10 times with different randomly selected initial conditions.

The quality of the forecast outbreak characteristics were derived from the ensemble mean trajectory and compared with observed outcomes. A forecast was deemed accurate if: (1) it peaked within ±1 week of the observed peak of infectious mosquitoes; (2) the maximum mosquito infection rate was within ±25% of the observed peak infection rate; (3) the total number of infectious

mosquitoes over the entire season was within ± 25% of the observed; and (4) the total number of human cases over the entire season was within ± 25% or ± 1 case of the total number of reported cases, whichever was larger. As an additional analysis, forecasts were examined across all years. All forecasts with the same lead were grouped and the fraction of accurate forecasts was quantified. For more details on the generation and analysis of these retrospective forecasts see the Supplementary Material.

**Code availability.** The code that support the findings of this study are available from the corresponding author upon request.

**Data availability.** Weekly human cases that support the findings of this study are available from ArboNET[30]. Mosquito observations from Suffolk County are available from Suffolk County Department of Health Services. All other data supporting the findings of this study are available from the corresponding author upon request.

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

## Acknowledgements

We are grateful to the staff of the Suffolk County Arthropod-Borne Disease Laboratory, the Division of Vector Control and the New York State Department of Health Arbovirus Laboratory for assistance in mosquito and arboviral surveillance efforts and viral analysis of the mosquito samples during the years of this study. We are also grateful to Paul Geery and the staff of the Desplaines Valley Mosquito Abatement District, Shamika Smith and the rest of the staff of the city of Chicago's arboviral surveillance efforts and Kelly Bemis and the Cook County health department. Zachary Schneider for contacting mosquito abatement districts and obtaining data. We also thank Jennifer Lehman and the CDC Division of Vector-Borne Infectious Diseases who provided us with weekly human WNV case data. This work was supported by US NIH grants GM100467 and ES009089, NIEHS training grant T32ES023770, and Defense Threat Reduction Agency contract HDTRA1-15-C-0018.

## Author contributions

N.B.D. and J.S designed the model, performed the analysis, interpreted the results and wrote the manuscript. E.L. and S.R.C. helped with data collection and interpretation and provided comments to the written manuscript.

## Additional information

**Competing financial interests:** J.S. discloses partial ownership of SK Analytics. The remaining authors declare no competing financial interests.

