## [Peer review file · Nature Communications]

Reviewers' comments:

Reviewer #1 (Remarks to the Author):

A. The authors designed a WNV model and methodology for using mosquito infection rates and reported human cases along with data assimilation methods to forecast WNV in real time during a season. Although the resulting forecasts were not always accurate, the model did quite well at predicting the timing of the mosquito infection peak a few weeks before the peak occurred. Once the mosquito infection peak occurred, the model also started predicting the total number of WNV human cases quite accurately. Since most human WNV cases are reported several weeks after the mosquito infection peak, the model is useful even though not particularly accurate before the peak occurs. Most importantly, this paper is a start towards true forecasting of WNV in the future in real time.

B. The methods and data usage are novel for WNV. Real-time forecasting in WNV (or any disease other than seasonal flu, really) is basically non-existent currently but could be revolutionary in terms of informing the public and policy makers about risk and use of mitigation.

C. The quality of the data and validity of the approach are good apart from a mistake in the ODE model form that is discussed in (F). One minor aspect of the model that should be addressed is the authors do not include an incubating 'E' compartment for mosquitoes, which could be important since that period is on the same order as the mosquito's lifespan (around 10 days). Also, this time adds to serial generation time, which can change the timing of, e.g. the peak. Please talk briefly about why this was not included in the model and how inclusion may change results.

Presentation of results is pretty good. It would be nice to have a figure that explicitly shows how far ahead of time the model is predicting total human cases accurately (e.g. in Discussion the authors claim that "accurate forecasts of total human WNV cases were generated up to 10 weeks prior to the end of an outbreak", but I can't tell how helpful this will be. What percent of human cases have been reported already by the time the model accurately predicts the total number of cases?).

D. Use of statistics appears to be appropriate and uncertainty is addressed in the forecasts by showing 25th to 75th and 10/90th percentiles. It would be helpful if the authors presented information about sensitivity of forecasts to the initial conditions and seeding of infected mosquitoes. Sampling the parameters only 100 times seems a bit low--would it be too computationally expensive to use more samples (Supplementary Material lines 237-250)? Similarly, are 300 runs for the EAKF sufficient? Have you explored this to make sure it is converging?

E. The general methods used are robust and if used consistently should be reliable. The baseline ODE model used may not be the best choice, particularly with use of the contact rate, β (see F).

F. For compartmental models, the force of infection term for mosquitoes is the product of the number of bites on birds per mosquito per unit time \times probability that it bites an infected bird \times probability transmission occurs. Similarly the force of infection for birds is the product of number of bites by a mosquito per bird per unit time \times probability a bite is from an infected mosquito \times probability transmission occurs.

The authors assume (by using β in both bird and mosquito force of infection) that the number of bites per mosquito per unit time \times probability of transmission of the bird is the same as the number of bites per bird per unit time \times probability of infection of the mosquito. This is almost certainly not true. Each mosquito only bites once every few days, while birds receive multiple bites per day.

However, this is relatively easy to fix:

-keep the mosquito force of infection the same as $\beta(t) * I_B/N_B$. This assumes that every mosquito gets a fixed number of bites per unit time

-change the bird force of infection to $\beta(t)N_M/N_B * I_M/N_M$. This assumes that the number of bites per bird per day are the total number of mosquito bites per day divided by the number of birds.

See M. J. Wonham, M. A. Lewis, J. Renc lawowicz and P. Van den Driessche, Transmission assumptions generate contradicting predictions in host-vector disease models: A case study in West Nile Virus, *Ecology Letters*, 9 (2006), 706-725 for details about this.

G. It would be very helpful for the authors to provide the distributions used for each parameter along with references for the choice of those distributions/ranges. A general better description of how the IC's and parameter distributions were chosen by the authors would be helpful for generalization/reproducibility.

H. The text is clear and well-written and the conclusions are appropriate for the output. Abstract is clear and appropriate.

Reviewer #2 (Remarks to the Author):

A. Using a relatively simple SIR modeling framework, the authors combine two different data streams to produce retrospective ensemble forecasts of the WNV outbreak in Long Island, NY between 2001 and 2014.

B. The work is of great interest as it demonstrates the feasibility of a real-time forecasting system for WNV. Their model is able to produce predictions with enough lead time to be "useful" for public health professionals.

C. The model and results were well presented. In particular Figure 3 did a very good job of explaining the forecasts and their increasing accuracy.

D. The treatment of uncertainties was appropriate.

E. My primary concern with this work is on the author's claim of generalizability. The modeling of WNV transmission is made exceptionally complex due to the presence of multiple hosts and vectors. Each of these has their own susceptibility (see, e.g., Turell 2001, Wheeler 2009). The host feeding preferences for these mosquitoes also varies (Molaei 2006 [Already discussed and referenced in MS]). Different strains of WNV also impact this differential susceptibility in hosts (Braut 2004). Vertical transmission can occur (Turell 2001) and over-wintering through diapause allows the usually short-lived mosquitoes to impact subsequent transmission seasons (Nasci 2001). Substantial heterogeneities (Kilpatrick 2006) greatly impact the applicability of any "well-mixed" transmission model. Finally, and perhaps most importantly for the purposes of predicting "when" within a season an outbreak occurs (as opposed to "if"), many of the most susceptible hosts in a region may be migratory birds that either bring the pathogen with them to the region, take the pathogen from the region, or miss the outbreak entirely. The model presented here ignores all of these complexities and is still able to do a reasonably accurate job of prediction in this particular location (side note: while some of the above complexities are mentioned in the manuscript, many are ignored and should be discussed). My primary concern with this work is its generalizability to other locations, in particular those locations where host composition is made a complex function of time due to migratory patterns.

F. While it is the classic "reviewer's experiment", I feel that this work would have significantly greater impact if the same approach was repeated and shown to be applicable in a new location that was somewhat epidemiologically different from the one already modeled. Perhaps a region where transmission was greatly influenced by hosts that migrate?

G. There are some complexities of WNV transmission that have not been mentioned (and thus associated references that are missing).

H. The text was adequately clear and lucid.

Reviewers' comments:

Reviewer #1 (Remarks to the Author):

A. The authors designed a WNV model and methodology for using mosquito infection rates and reported human cases along with data assimilation methods to forecast WNV in real time during a season. Although the resulting forecasts were not always accurate, the model did quite well at predicting the timing of the mosquito infection peak a few weeks before the peak occurred. Once the mosquito infection peak occurred, the model also started predicting the total number of WNV human cases quite accurately. Since most human WNV cases are reported several weeks after the mosquito infection peak, the model is useful even though not particularly accurate before the peak occurs. Most importantly, this paper is a start towards true forecasting of WNV in the future in real time.

B. The methods and data usage are novel for WNV. Real-time forecasting in WNV (or any disease other than seasonal flu, really) is basically non-existent currently but could be revolutionary in terms of informing the public and policy makers about risk and use of mitigation.

C. The quality of the data and validity of the approach are good apart from a mistake in the ODE model form that is discussed in (F). One minor aspect of the model that should be addressed is the authors do not include an incubating 'E' compartment for mosquitoes, which could be important since that period is on the same order as the mosquito's lifespan (around 10 days). Also, this time adds to serial generation time, which can change the timing of, e.g. the peak. Please talk briefly about why this was not included in the model and how inclusion may change results.

Response:

This is a great point. We tested the forecast system by adding an exposure compartment to the model to see if it would increase forecast accuracy. Using the following model structure:

$$\frac{dS_M}{dt} = \mu_M N_M - \beta(t) S_M \frac{I_B}{N_B} - \mu_M S_M - \alpha S_M \quad [1]$$

$$\frac{dE_M}{dt} = \beta(t) S_M \frac{I_B}{N_B} - \phi_M E_M - \mu_M E_M \quad [2]$$

$$\frac{dI_M}{dt} = \phi_M E_M - \mu_M I_M + \alpha S_M \quad [2]$$

$$\frac{dS_B}{dt} = -\beta(t) I_M \frac{S_B}{N_B} \quad [3]$$

$$\frac{dI_B}{dt} = \beta(t) I_M \frac{S_B}{N_B} - \frac{I_B}{\delta_B} \quad [4]$$

$$\frac{dI_H}{dt} = \text{Poisson}(\eta I_M) \quad [5]$$

where S_M is the number of susceptible mosquitoes, μ_M is the mosquito birth and death rate, N_M is the mosquito population, t is time in days, $\beta(t)$ is the contact rate, or probability of transmission between birds and mosquitoes, α is the rate of WNV seeding prior to day 200, E_M is the number of mosquitoes exposed to WNV but not yet infectious, ϕ_M is the extrinsic incubation period, I_M is the number of infected mosquitoes, N_B is the bird population, I_B is the number of infected birds, S_B is the number of susceptible birds in the population, δ_B is the recovery rate of birds, I_H is the number of infected humans, and η is a scaling factor representing the probability of spillover transmission to humans from mosquitoes. This form is similar to the model used in the paper but has one additional parameter (ϕ_M) and one additional variable (E_M).

The new model form was able to freely simulate observed infectious mosquitoes and human cases, Fig. C1. When combined with the EAKF, the system was also able to constrain infectious humans quite well but did not constrain infectious mosquitoes as well as the system using the model without the additional compartment (Fig C1). This error is most likely due to a more limited optimization of $\beta(t)$, which is generally higher than the truth, Fig. C2, and results in greater numbers of infected mosquitoes later in the season and less accurate forecast, Fig. C3.

Fig C1. The top plot is average weekly observations (2001-2014, red *) and free model simulation of exposed and infectious mosquito rates which are the blue and red lines, respectively. The second and third plots are time series results of the posterior ensemble mean for the observed state space, I_M and I_H , generated using the compartmental model-EAKF framework, and truth, cyan colored dotted line. The 1,000 simulations are represented in the box and whiskers, which show the median (red horizontal line), 25th and 75th percentiles (box boundaries), the whiskers mark the highest and lowest values within 1.5 times the inter quartile range of the box boundaries and outliers (red +). The SEI-SI-I compartmental model-EAKF system does not constrain the number of infectious mosquitos as well as the SI-SI-I model-EAKF system, particularly during the later half of the outbreak.

Fig C2. Time series results of the posterior ensemble mean for parameter $\beta(t)$ generated using the compartmental model-EAKF framework, and truth, cyan colored dotted line. The box and whiskers shows show the median (red horizontal line), 25th and 75th percentiles (box boundaries), the whiskers mark the highest and lowest values within 1.5 times the inter quartile range of the box boundaries and outliers (red +). The SEI-SI-I compartmental model-EAKF system does not constrain beta as well as the SI-SI-I model-EAKF system. Late in the year infected mosquitoes are higher than observed.

Fig C3. Results for 2001-2014 retrospective forecasts. Shown are the fraction of forecasts accurate as a function of lead week for the metrics human WNV cases (blue), peak timing (week of peak mosquito infection rates, orange), peak infection rate (yellow), and total infectious mosquitoes (purple). A forecast was deemed accurate if: 1) peak timing was within ± 1 week of the observed peak of infectious mosquitoes; 2) peak infection rate was within $\pm 25\%$ of the observed peak infection rate; 3) total infectious mosquitoes were within $\pm 25\%$ of the observed; and 4) human WNV cases were within $\pm 25\%$ or ± 1 case of the total number of reported cases, whichever was larger. Note that for all metrics lead week is shown with respect to the week of peak mosquito infection. The high number of infected mosquitoes in the later half of the outbreak appears to decrease forecast accuracy of the total human cases and total infectious mosquitoes.

One of the critical decisions we had to make when building the model for WNV forecast was to decide which processes to include and which to exclude. Inclusion of too many processes results in a high-dimensional model structure, which, given the limited observational data streams available, may be difficult to optimize. On the other hand, a model that is too simple will not contain sufficient dynamics to generate a characteristic WNV outbreak in free simulation.

The model we chose for the present work lies in the middle—neither too complex nor too simple. It is capable of producing a realistic outbreak of WNV in

free simulation, yet is sufficiently parsimonious to permit state variable and parameter estimation with the EAKF and accurate forecast of future outcomes. Exclusion of an incubating 'E' compartment was a choice we made in favor of model parsimony. While inclusion of such a compartment would be more realistic, as the above plots demonstrate, the additional complexity appears to undermine optimization and forecast accuracy given available data streams. In the future, particularly should richer observations come available, it may be worthwhile to incorporate such processes explicitly in the model framework.

We have added discussion of the above issues to the revised the manuscript.

Presentation of results is pretty good. It would be nice to have a figure that explicitly shows how far ahead of time the model is predicting total human cases accurately (e.g. in Discussion the authors claim that "accurate forecasts of total human WNV cases were generated up to 10 weeks prior to the end of an outbreak", but I can't tell how helpful this will be. What percent of human cases have been reported already by the time the model accurately predicts the total number of cases?).

Response:

We have added a figure (Figure S8, shown below) to the supplementary materials, which shows the number of weeks forecasts were accurate prior to the end of the outbreak. Accurate forecasts of total human WNV cases were generated on average 5.4 weeks prior to the end of an outbreak.

Fig. S8. The number of weeks of accurate forecast of total human WN cases prior to week 42, the typical last week of the outbreak. For 2003 the outbreak ended in week 45, so weeks prior to week 45 are presented. Each bar is the average of 10 ensemble forecasts (see Methods), hence the values are continuous not integer.

D. Use of statistics appears to be appropriate and uncertainty is addressed in the forecasts by showing 25th to 75th and 10/90th percentiles. It would be helpful if the authors presented information about sensitivity of forecasts to the initial conditions and seeding of infected mosquitoes. Sampling the parameters only 100 times seems a bit low--would it be too computationally expensive to use more samples (Supplementary Material lines 237-250)? Similarly, are 300 runs for the EAKF sufficient? Have you explored this to make sure it is converging?

Response:

We ran each ensemble seasonal forecast 10 times, each time with different randomly chosen initial conditions and seeding. This is now described in the revised manuscript. The results presented in the manuscript (both the original and revised) show the average accuracy across all 10 of these forecasts (i.e. 10 iterations and 14 seasons of weekly forecasts). The results among the iterations

are very similar, showing nominal sensitivity to initial conditions and seeding.

Use of 100 replicates for synthetic tests is consistent with our previous efforts. We have found good agreement among the replicates in these tests. In our previous work with compartmental models, we have explored the effect of ensemble size on model optimization (see Yang et al., 2014, PLOS Comp Biol). We found an ensemble of 300 is adequate for the EAKF.

Figs D1 to D3 are box and whisker plots showing the variability among 100 different posterior ensemble mean estimates for one synthetic truth. Each ensemble member was initialized with a constant total population and randomly selected parameters (from the uniform distributions given in the text). The limited variability between posterior ensemble means shows model convergence and nominal sensitivity to initial conditions and seeding.

Fig. D1. Time series results of the posterior ensemble mean for the observed state space, I_M and I_H , generated using the compartmental model-EAKF framework, and truth, cyan colored dotted line. The 100 simulations are represented in the box and whiskers, which show the median (red horizontal line), 25th and 75th percentiles (box boundaries), the whiskers mark the highest

and lowest values within 1.5 times the inter quartile range of the box boundaries and outliers (red +).

Fig. D2. Time series results of the posterior ensemble mean for the unobserved state space, S_M , S_B and I_B , generated using the compartmental model-EAKF framework, and truth, cyan colored dotted line. The box and whiskers show the median (red horizontal line), 25th and 75th percentiles (box boundaries), the whiskers mark the highest and lowest values within 1.5 times the inter quartile range of the box boundaries and outliers (red +).

Fig. D3. Time series results of the posterior ensemble mean for parameters μ_M , $\beta(t)$, δ_B , and η , generated using the compartmental model-EAKF framework, and truth, cyan colored dotted line. The box and whiskers shows show the median (red horizontal line), 25th and 75th percentiles (box boundaries), the whiskers mark the highest and lowest values within 1.5 times the inter quartile range of the box boundaries and outliers (red +).

E. The general methods used are robust and if used consistently should be reliable. The baseline ODE model used may not be the best choice, particularly with use of the contact rate, beta (see F).

Response:

We changed the baseline ODE model per the comments in F.

F. For compartmental models, the force of infection term for mosquitoes is the product of the number of bites on birds per mosquito per unit time x probability that it bites an infected bird x probability transmission occurs. Similarly the force of infection for birds is the product of number of bites by a mosquito per bird per unit time x probability a bite is from an infected mosquito x probability transmission occurs.

The authors assume (by using beta in both bird and mosquito force of infection) that the number of bites per mosquito per unit time x probability of transmission of the bird is the same as the number of bites per bird per unit time x probability of infection of the mosquito. This is almost certainly not true. Each mosquito only bites once every few days, while birds receive multiple bites per day.

However, this is relatively easy to fix:

-keep the mosquito force of infection the same as $\beta(t) * I_B/N_B$. This assumes that every mosquito gets a fixed number of bites per unit time

-change the bird force of infection to $\beta(t)N_M/N_B * I_M/N_M$. This assumes that the number of bites per bird per day are the total number of mosquito bites per day divided by the number of birds.

See M. J. Wonham, M. A. Lewis, J. Renc lawowicz and P. Van den Driessche, Transmission assumptions generate contradicting predictions in host-vector disease models: A case study in West Nile Virus, Ecology Letters, 9 (2006), 706{725 for details about this.

Equations from M. J. Wonham, M. A. Lewis, J. Renc lawowicz and P. Van den Driessche, Transmission assumptions generate contradicting predictions in host-vector disease models: A case study in West Nile Virus, Ecology Letters, 9 (2006), 706{725 for details about this.

Response:

This is a fantastic point. We have changed the model structure to the following equations:

$$\frac{dS_M}{dt} = \mu_M N_M - \beta(t) S_M \frac{I_B}{N_B} - \mu_M S_M - \alpha S_M \quad [1]$$

$$\frac{dI_M}{dt} = \beta(t) S_M \frac{I_B}{N_B} - \mu_M I_M + \alpha S_M \quad [2]$$

$$\frac{dS_B}{dt} = -\beta(t) I_M \frac{S_B}{N_B} \quad [3]$$

$$\frac{dI_B}{dt} = \beta(t) I_M \frac{S_B}{N_B} - \frac{I_B}{\delta_B} \quad [4]$$

$$\frac{dI_H}{dt} = \text{Poisson}(\eta I_M) \quad [5]$$

Using this form, we re-ran the initial tests to validate that beta behaved as found previously, and it did (see figure S2 of the revised manuscript). We also re-ran the synthetic tests to determine whether EAKF optimization was similarly functional, and it was (see Figures S3A-S3D). Forecasts with this model, which is now presented in the revised manuscript, are quite comparable to that from the original manuscript. The new forecast is slightly better at predicting human cases. In the revised manuscript at 0, 1 and 2 weeks past peak predicted mosquito infection rates, the forecasts of total human WNV cases were accurate 100%, 65% and 72% of the time, respectively, while in the previous manuscript predicted 64%, 60% and 69%, respectively.

G. It would be very helpful for the authors to provide the distributions used for each parameter along with references for the choice of those distributions/ranges. A general better description of how the IC's and parameter distributions were chosen by the authors would be helpful for generalization/reproducibility.

Response: We have made some changes to the manuscript to provide more information on how the initial conditions and parameters were determined. Some were selected from the literature and others were derived from the observational data and the EAKF. In particular, we have specified the following:

The initial conditions for each ensemble run were selected randomly from uniform distributions. Mosquito expected lifespan (Wonham et al. 2004 and Hartley et al. 2012) and bird duration of infection (Komar et al. 2003) were selected from the literature, while $\beta(t)$ and η were derived during synthetic testing of the system.

H. The text is clear and well-written and the conclusions are appropriate for the output. Abstract is clear and appropriate.

Response: We appreciate the positive feedback.

Reviewer #2 (Remarks to the Author):

A. Using a relatively simple SIR modeling framework, the authors combine two different data streams to produce retrospective ensemble forecasts of the WNV outbreak in Long Island, NY between 2001 and 2014.

B. The work is of great interest as it demonstrates the feasibility of a real-time forecasting system for WNV. Their model is able to produce predictions with enough lead time to be "useful" for public health professionals.

C. The model and results were well presented. In particular Figure 3 did a very good job of explaining the forecasts and their increasing accuracy.

D. The treatment of uncertainties was appropriate.

Response: We appreciate the positive feedback.

E. My primary concern with this work is on the author's claim of generalizability. The modeling of WNV transmission is made exceptionally complex due to the presence of multiple hosts and vectors. Each of these has their own susceptibility

(see, e.g., Turell 2001, Wheeler 2009). The host feeding preferences for these mosquitoes also varies (Molaei 2006 [Already discussed and referenced in MS]). Different strains of WNV also impact this differential susceptibility in hosts (Brault 2004). Vertical transmission can occur (Turell 2001) and over-wintering through diapause allows the usually short-lived mosquitoes to impact subsequent transmission seasons (Nasci 2001). Substantial heterogeneities (Kilpatrick 2006) greatly impact the applicability of any "well-mixed" transmission model. Finally, and perhaps most importantly for the purposes of predicting "when" within a season an outbreak occurs (as opposed to "if"), many of the most susceptible hosts in a region may be migratory birds that either bring the pathogen with them to the region, take the pathogen from the region, or miss the outbreak entirely. The model presented here ignores all of these complexities and is still able to do a reasonably accurate job of prediction in this particular location (side note: while some of the above complexities are mentioned in the manuscript, many are ignored and should be discussed). My primary concern with this work is its generalizability to other locations, in particular those locations where host composition is made a complex function of time due to migratory patterns.

Response: As mentioned above in response to Reviewer 1, one of the critical decisions we had to make when building the model for WNV forecast was to decide which processes to include and which to exclude. Inclusion of too many processes results in a high-dimensional model structure, which, given the limited observational data streams available, may be difficult to optimize. On the other hand, a model that is too simple will not contain sufficient dynamics to generate a characteristic WNV outbreak in free simulation and thus will not produce accurate forecasts.

The model we chose for the present work lies in the middle—neither too complex nor too simple. It is capable of producing a realistic outbreak of WNV in free simulation, yet is sufficiently parsimonious to permit state variable and parameter estimation with the EAKF and accurate forecast of future outcomes. In the future, if richer, more detailed observations are available, we may be able to utilize a more complex model that includes some of the processes the reviewer mentions. We now discuss these issues in the revised manuscript.

We constructed the model to be simple and generalizable. To test this generalizability, we have now applied the model-inference forecast system to one additional location, Cook County, Illinois. Forecast accuracy was similar to that found for Suffolk County. In particular, we found that forecasts of infectious mosquito peak timing were accurate >55% of the time with a six-week prediction lead. Forecasts of the peak infectious mosquito number were not as accurate as for Suffolk County with long lead times but were >40% accurate when the predicted lead-time was at the peak and >69% past the peak. For the total number of infected mosquitoes 45% of forecasts were accurate one week prior to the predicted peak, 50% were accurate at the predicted peak and 87% were accurate one week past the peak. At 0, 1 and 2 weeks past peak predicted

mosquito infection rates, the forecasts of total human WNV cases were accurate 40%, 74% and 69% of the time, respectively. On average, only 18% of human cases are reported at the peak, 34% of human cases are reported one week past the peak and 46% are reported 2 weeks past the peak. Human cases were accurately forecast 5 to 11 weeks prior to the end of the outbreak (Fig. S11). These results are now presented in Supplementary Information. Note that it is the EAKF optimization, which in part compensates for model mis-specification that allows this relatively simple construct to produce accurate forecasts in locations where WNV, host and vector dynamics differ.

F. While it is the classic "reviewer's experiment", I feel that this work would have significantly greater impact if the same approach was repeated and shown to be applicable in a new location that was somewhat epidemiologically different from the one already modeled. Perhaps a region where transmission was greatly influenced by hosts that migrate?

Response:

As noted above, we constructed the model to be simple and generalizable. We tested the generalizability by applying the model-inference forecast system to Cook County, Illinois. The results from Cook County show promise that the model can adapt to other regions of the county. This is because EAKF optimization in part compensates for model mis-specification and allows this relatively simple construct to produce accurate forecasts in locations where WNV, host and vector dynamics differ. Thus, the model-EAKF systems does present a structure from which one can build as more robust data sets become available. We now discuss these issues in the revised manuscript.

G. There are some complexities of WNV transmission that have not been mentioned (and thus associated references that are missing).

Response:

We expanded the discussion section to include the complexities of WNV transmission. We now discuss some of the processes not included in the core dynamic model, including within county spatial heterogeneity, migratory patterns of birds, varying susceptibility among multiple hosts and vectors, the impact of varying strains of WNV, vertical transmission, mosquitoes that over winter, and the extrinsic incubation period.

H. The text was adequately clear and lucid.

Response: We appreciate the positive feedback.

REVIEWERS' COMMENTS:

Reviewer #1 (Remarks to the Author):

The authors did a great job of addressing concerns and questions in the revision. In particular, they examined the SEI version of the mosquito model and updated the transmission terms in their original model as suggested. They provided more clarifying information about parameters and computational methods and convergence.

Reviewer #2 (Remarks to the Author):

I feel that the authors have adequately addressed my previous concerns, and that the revised manuscript does an excellent job presenting their analysis and conclusions.

REVIEWERS' COMMENTS:

Reviewer #1 (Remarks to the Author):

The authors did a great job of addressing concerns and questions in the revision. In particular, they examined the SEI version of the mosquito model and updated the transmission terms in their original model as suggested. They provided more clarifying information about parameters and computational methods and convergence.

Response: We appreciate the positive feedback.

Reviewer #2 (Remarks to the Author):

I feel that the authors have adequately addressed my previous concerns, and that the revised manuscript does an excellent job presenting their analysis and conclusions.

Response: We appreciate the positive feedback.